# Validation of DOE Factorial/Taguchi/Surface Response Models of Mechanical Properties of Synthetic and Natural Fiber Reinforced Epoxy Matrix Hybrid Material

**DOI:** 10.3390/polym16142051

**Published:** 2024-07-18

**Authors:** Oscar G. Toapanta, Juan Paredes, Manuel Meneses, Gabriela Salinas

**Affiliations:** 1Escuela de Posgrado, Universidad Nacional de Chimborazo, Riobamba 060110, Ecuador; oscartoapantaambjlm@gmail.com (O.G.T.); ameneses@unach.edu.ec (M.M.); 2Escuela Doctorado, Universidad Nacional de Ingeniería (UNI), Lima 15333, Peru; 3Campus Benjamín Araujo, Instituto Superior Tecnológico Pelileo, Patate 180550, Ecuador; 4GI3M—Grupo de Investigación e Innovación en Ingeniería Mecánica, Universidad Técnica de Ambato, Ambato 180104, Ecuador; vg.salinas@uta.edu.ec; 5Escuela Internacional de Doctorado (EIDUNED), Universidad Nacional de Educación a Distancia (UNED), 28040 Madrid, Spain; 6Facultad de Ingeniería Civil y Mecánica, Universidad Técnica de Ambato, Ambato 180104, Ecuador; 7Facultad de Ciencias de la Salud, Universidad Técnica de Ambato, Ambato 180105, Ecuador

**Keywords:** hybrid materials, natural fibers, factorial model, Taguchi method, response surface methodology (RSM), mechanical testing and mechanical properties

## Abstract

A validation of the factorial, Taguchi and response surface methodology (RSM) statistical models is developed for the analysis of mechanical tests of hybrid materials, with an epoxy matrix reinforced with natural Chambira fiber and synthetic fibers of glass, carbon and Kevlar. These materials present variability in their properties, so for the validation of the models a research methodology with a quantitative approach based on the statistical process of the design of experiments (DOE) was adopted; for which the sampling is in relation to the design matrix using 90 treatments with three replicates for each of the study variables. The analysis of the models reveals that the greatest pressure is obtained by considering only the source elements that are significant; this is reflected in the increase in the coefficient of determination and in the predictive capacity. The modified factorial model is best suited for the research, since it has an R^2^ higher than 90% in almost all the evaluated mechanical properties of the material; with respect to the combined optimization of the variables, the model showed an overall contribution of 99.73% and global desirability of 0.7537. These results highlight the effectiveness of the modified factorial model in the analysis of hybrid materials.

## 1. Introduction

The generation and obtaining of new materials arises from the need to increase the physical, chemical and mechanical properties of the elements that already exist in nature [1,2]. Thus, hybrid materials arise from the combination of simple materials with different characteristics, forming phases of a matrix and a reinforcement. The matrix gives the composite ductility, toughness and transmits the induced stresses to the reinforcements so that they support most of the applied force [3]. The properties of a hybrid material will depend on the elements that constitute it, their distribution and interaction; obtaining appropriate elements for each application [4]. Its classification is based on the shape or nature of the constituents and the size of the dispersed phase; offering advantages such as high strength and stiffness relative to its weight, the ability to adapt the material to the required stresses through anisotropy and versatility in the design of complex shapes [5,6,7].

In recent years, the use of composite materials has become widespread worldwide in all aspects of human life, diversified and with greater volumes of consumption [8], with synthetic fibers such as thermoplastics and thermosets reinforced with carbon fibers, aramid (Kevlar), glass fibers, natural fibers and others [9,10,11]. The increase in environmental pollution has generated laws for its care and protection; one option to help with this problem is the ecological or renewable hybrid materials that are called biocomposites, which are obtained by the mixture of biopolymers reinforced with natural fibers of vegetable origin that combined with other elements improve their properties and applications, and causing them to be highly competitive [12]. The use of synthetic materials stimulates environmental concerns, and researchers around the world are reacting effectively to environmental concerns by moving towards biodegradable and sustainable materials [13,14]. There are several advantages to switching from synthetic fibers to natural fibers, such as easy availability, biodegradability, low cost, low density, minimal health risks and environmental friendliness [15,16]. Natural fiber composites [17] (such as jute, flax, sisal and abaca fibers) have a wide range of applications, and stand out in the automotive industry in different components; this is due to their lightness and reasonable resistance, contributing to environmental sustainability by reducing the dependence on synthetic materials [18,19]. Banana, jute and kenaf fibers are used in civil construction; and bamboo and ramie in sports equipment and clothing [20,21]. Table 1 shows some applications of natural fiber composites [22].

There are several researches into different natural fibers such as coconut fiber, jute, sisal and flax—that are being used in a large number of applications due to their advantages in terms of mechanical properties; since they improve tensile and flexural stresses, and their weight is lower than other traditional materials [23,24]. For example, it was found that the inclusion of jute and sisal fibers used as reinforcement in the composite material increases the mechanical properties such as modulus of elasticity, tensile strength and impact strength [25]. There were also good results in tensile and flexural stress with the Cumare palm fiber (Astrocaryum Chambira), which is under analysis in this research [26]. However, the design and optimization of these materials poses unique challenges, related to the inherent variability of the properties of the natural fibers and their interaction with the polymer matrix [27]. The objective of the present research is to validate the application of design of experiments (DOE) models specifically the factorial model, Taguchi and response surface methodology (RSM) in the mechanical properties of the hybrid material of epoxy matrix reinforced with synthetic and natural fibers; through which it will be possible to establish its effectiveness, accuracy and applicability in the prediction of its mechanical properties and how to optimize them. Furthermore, the use of hybrid materials with natural fibers will have a significant impact on the reduction of environmental pollution.

The DOE uses a systematic process that begins with the identification of the objectives of the experiment and then establishes the factors and levels of study; this allows the selection of the appropriate design of experiments, with which the experimental matrix is established in relation to the combinations of factors. Through these, the experimentation is carried out and then the recording, analysis and interpretation of the data are obtained, according to the statistical techniques established for each model such as the analysis of variance (ANOVA) and the signal/noise ratio (S/N) that allow the evaluation of the linear and quadratic effects and their interactions according to the model of design of experiments that in the present work is analyzed: the factorial, Taguchi and RSM. It is essential to perform this analysis due to there being little research on the comparative evaluation of these statistical models, such as the one carried out on the analysis of the influence of factorial designs in the characterization of complex systems [28], and the one carried out on ethanol production comparing the Taguchi and RSM methodology [29].

There are studies that analyze the aforementioned models individually for hybrid materials with natural fiber reinforcement, such as the research on the improvement of the mechanical properties of the composite material with coconut fiber using the full factorial methodology [30]. There have also been studies performed on: the influence of the orientation of the sisal reinforcement fibers in the biopolymer composite using a factorial design [31]; as well as other factorial studies on the search for better composite materials with natural fibers such as flax, jute, sisal and ichu [32]. The Taguchi method with L4 orthogonal array has been used to determine the best bending stress in bamboo–banana fiberglass composite [33]; and research on optimal production parameters of polymer matrix and fiber biocomposites from kenaf and jute fabric with Taguchi L27 matrix [34] has been conducted; also, investigations on the physicochemical and tensile properties by Taguchi L16 using natural Date Palm rachis fibers as reinforcement [35] have been done, as have studies with woven natural fiber reinforcement [36]. With respect to the response surface methodology (RSM), we have the study using Composite Central Design (CCD) with 27 cases in the modelling of the mechanical properties of the abaca fiber reinforced composite [37]. Optimization of stress and tensile modulus in flax fiber-reinforced material using the RSM and Box–Behnken techniques, with a matrix of 17 treatments [38] has been studies and so has the improvement of material properties by adding fibers such as flax, jute, sisal, bamboo and kenaf with RSM [39]. In relation to the above, the research hypothesis is the evaluation of the design of the factorial, Taguchi and RSM experiment models used for the analysis of the results of the mechanical tests of the hybrid material with natural fibers. This will allow us to determine the accuracy and predictive capacity of the mechanical properties of the material.

## 2. Materials and Methods

The research is experimental, since the variables of interest will be directly manipulated to obtain the specimens of the hybrid material; the data will be obtained experimentally through mechanical tests on the hybrid material with natural Cumare palm fiber (Astrocaryum Chambira) [40]. This will allow evaluation of precision and predictive capacity through a quantitative explanatory approach of the statistical models of experimental design (DOE) established for the analysis. Figure 1 shows the process to obtain the hybrid material with natural Chambira fiber [40,41].

The population is composed of the mechanical tests that can be carried out on composite materials with natural fibers of Chambira, which was selected for its good properties in relation to other types of natural fibers. Table 2 below shows the physical and mechanical properties of some natural fibers used as reinforcement for hybrid materials [3,22].

For the research, sampling based on the DOE design of experiment is used in the development of the epoxy polymeric epoxy-based hybrid material KDA-HI [42], with synthetic fiber and natural Chambira fiber reinforcement [40,41]; this was carried out by means of the considerations of the matrix arrangements established for the methodologies of the factorial, Taguchi and RSM design of experiments models. These are based on the combination of the three continuous quantitative variables, which for the investigation is the orientation of the natural fibers in two different layers—each one of these with the levels of −45, 0 and 45 degrees—these orientations are related to the resistance that can be obtained in the hybrid materials and the selected values are those that have greater influence on the relation between the tension and deformation of these materials [33,43]; and the third quantitative variable is the drying temperature of the process, which was established based on the neutral curing temperature of 90 °C; and that of 60 °C and 120 °C in relation to other investigations, where excellent mechanical properties are obtained in hybrid materials [44,45]. The arrangements analyzed were the full factorial design 3^3^ obtaining 27 cases as well as the Taguchi L27 arrangement [34,37]; a Taguchi L9 arrangement generates 9 cases [36]; using the response surface methodology by the CCD Composite Central Design technique yields 20 cases [46]; and using the Box–Behnken technique for the same methodology generates 15 cases [47,48]. The objective of the DOE design of experiments is to minimize economic and time resources. Therefore, a suitable matrix arrangement was selected to guarantee the reliability of the mechanical properties of the composite material to be obtained, which was with a total of 15 cases. In addition, 2 qualitative variables were added; the first one is the type of synthetic fiber—glass fiber, carbon fiber and Kevlar fiber—since they have good mechanical properties such as tensile strength, modulus of elasticity and toughness, that contribute to generating an adequate hybrid material [49,50]. The other factor is the type of molding process of fusion molding and vacuum molding [51,52,53].

Therefore, the sampling for the present investigation is 15 × 3 × 2 × 3, which is detailed in 15 cases of the selected DOE design of experiments matrix, which is combined with the 3 types of synthetic fiber and 2 types of molding process of the hybrid material; obtaining 90 study treatments that combine the five independent variables of the investigation. To each treatment, 3 repetitions are applied; that is to say that the sample is 270 specimens for each of the response variables that are the mechanical properties of the hybrid material. These are separated into groups; the first group is made up of the tensile stress, tensile modulus of elasticity and percent elongation at failure, using the universal machine WAW600B and the American Society for Testing and Materials (ASTM) D3039 standard, which consists of the application of an axial force at two points until the material reaches its rupture; the other group is the flexural stress, modulus of elasticity at flexure and deflection using the ASTM D7264 standard that examines the vertical load applied at the center of the supported material; finally, the impact resistance property using the failure energy using the ASTM 5628 standard this is performed by holding the material sample and dropping a dart from a given height to measure the energy required to fracture it [54].

The analysis of the evaluation of DOE models will allow a more efficient and accurate optimization of composites reinforced with natural fibers. In materials design, therefore, the use of factorial models has proven to be a crucial statistical and mathematical tool to analyze the effect of various factors and their interactions. The following is the mathematical regression model for a factorial design with 5 independent variables, which is a function of their main effects and all possible iterations in a linear order, as shown in Equation (1) [32,55,56].
(1)$$y=\beta_{0}+\sum_{i=1}^{n}\beta_{i}x_{i}+\sum_{i=1}^{n}\sum_{j=2}^{n}\beta_{ij}x_{i}x_{j}+\sum_{i=1}^{n}\sum_{j=2}^{n}\sum_{k=3}^{n}\beta_{ijk}x_{i}x_{j}x_{k}+\cdots +\sum_{i=1}^{n}\sum_{j=2}^{n}\sum_{k=3}^{n}\sum_{l=4}^{n}\sum_{m=5}^{n}\beta_{ijklm}x_{i}x_{j}x_{k}x_{l}x_{m}+\varepsilon \;\;\;\;\;\;\;\;\;\;\;\;$$

The Taguchi methodology has established itself as an effective method in materials engineering, particularly for the design and refinement of natural fiber-reinforced composites. It stands out for its systematic and statistically based procedure for quality management, facilitating the improvement of the performance of composite materials and the reduction of their variability. The mathematical regression model with Taguchi’s methodology for the arithmetic mean of the variables and the signal/noise ratio is given in Equation (2), when it has no effects due to iterations [36]; and Equation (3), when linear iterations are considered [57,58].
(2)$$y=\beta_{0}+\sum_{i=1}^{n}\beta_{i}x_{i}+\varepsilon \;\;\;\;\;\;\;\;\;\;\;\;\;$$
(3)$$\;\;\;\;\;\;\;\;\;\;y=\beta_{0}+\sum_{i=1}^{n}\beta_{i}x_{i}+\sum_{i=1}^{n}\sum_{j=2}^{n}\beta_{ij}x_{i}x_{j}+\varepsilon \;\;\;\;\;\;\;\;\;\;\;\;\;$$

The surface design technique emerges as a sophisticated statistical resource for the optimization and modelling of complex systems in the field of composite materials, providing the ability to simultaneously evaluate various factors and their interactions. This method is particularly notable for its usefulness in experimentation with multiple variables and levels, providing a detailed understanding of the impact of these factors on the characteristics of composite materials. The mathematical regression model for RSM with 5 factors is of second order with the Box–Behnken technique for the continuous quantitative factors, and with linear interactions of its quantitative and qualitative variables as shown in Equation (4) [59,60,61].
(4)$$y=\beta_{0}+\sum_{i=1}^{n}\beta_{i}x_{i}+\sum_{i=1}^{n}\beta_{ii}x_{i}^{2}+\sum_{i=1}^{n}\sum_{j=2}^{n}\beta_{ij}x_{i}x_{j}+\varepsilon \;\;\;\;\;\;\;\;\;\;\;\;\;\;$$

## 3. Results

The summary of the data obtained from the mechanical properties of the hybrid material of epoxy matrix with synthetic and natural fiber is shown in Table 3; as indicated, it is based on a design of experiment (DOE) of 90 treatments with three repetitions for each one; generated by the three continuous quantitative independent variables combined with the two qualitative variables for each mechanical property, so the total number of data collected from the tests is 1890 with which the analysis and evaluation process is carried out.

### 3.1. Comparative Analysis of Design of Experiments Models

The analysis of the factorial, response surface methodology (RSM) and Taguchi models allows us to establish which is the most adequate for the design and optimization process of the composite material with natural fibers, through the maximization of its mechanical properties. Table 4 shows the statistical hypotheses, null (H_0_) and alternative (H_1_) used for the analysis and evaluation of the models in relation to the variables established for the research.

To start the analysis of the DOE models, the assumption of normality of the initial data of the mechanical properties was checked, most of which did not comply with the exception of deflection; so a Box–Cox transformation was performed, using a λ coefficient of 0.0 for tensile stress, tensile and flexural modulus of elasticity; and a λ value of −0.5 for elongation, flexural stress and failure energy; showing that all the properties comply with normality. Next, we have the evaluation of each of the design of experiments models that was performed by ANOVA analysis with the established statistical hypotheses, and also with the coefficients of determination of the models and their prediction to maximize the mechanical properties of the hybrid material.

#### 3.1.1. Factorial Model

The analysis and evaluation of the Factorial model was carried out for each of the study variables; first of all, all the factors and all the linear iterations were analyzed, which is shown in Table 5; and then a modification in the model was established; that is, to make an adjustment using the source elements that are significant in the initial ANOVA, the results of which are shown in Table 6.

#### 3.1.2. Taguchi Model

The initial Taguchi model performed for the mechanical properties of the hybrid material is a function of the study factors and without considering the iterations; the results of which are shown in Table 7, while the responses of the modified Taguchi model—which is obtained by adding the linear iterations of the combinations of two terms—are shown in Table 8.

#### 3.1.3. Response Surface Methodology Model (RSM)

For the analysis of the response surface methodology with the Box–Behnken technique, the model was established with second-order effects and with linear iterations of the source elements, for each property of the hybrid material; the results are in Table 9, and Table 10 shows the responses of the modified RSM, which is generated by adjusting the model, without considering the source elements that are not significant in the first analysis.

In all DOE models, the assumptions of normality, equality of variance and independence were verified with the standardized residuals of the model fulfilling to satisfaction for a reliability of 95%.

### 3.2. Optimization of the Mechanical Properties of Hybrid Material

A fundamental aspect to compare the analysis models is through the prediction values of the mechanical properties of the composite material in relation to the factors established to obtain them, as well as the global desirability parameter calculated for the multiple optimization process in the DOE factorial and RSM; in the case of the Taguchi technique, the prediction values are with respect to the analysis of the best values of the mean and the signal/noise ratio. The optimization was performed with the modified models that generate a better prediction, such as the modified DOE factorial and RSM models, while in Taguchi it is the model with iterations, as shown in Figure 2.

The results observed in Figure 2 of the multiple optimizations for the combined form maximization of the mechanical properties of the hybrid material with Chambira natural fiber were generated in the modified factorial DOE model, with an orientation of the natural fiber in layer 1 (C1) and layer 2 (C2) at −45 and 0 degrees correspondingly, at a temperature of 120 degrees Celsius (°C), with the infusion molding type and using the combination with the Kevlar synthetic fiber. For the modified response surface model, adequate optimization is obtained with similar parameters in the molding process and in the synthetic fiber by switching to a C1 orientation of −7.7 degrees and C2 of −5.0 degrees and with a temperature of 60 °C. While in the Taguchi design with equal iterations the best prediction is with the same process and synthetic fiber, changing the orientation of the natural fiber C1 and C2 to 0 degrees, and with a temperature of 90 °C. By means of ANOVA analysis of the composite desirability function, the contribution that each effect has in the DOE models of analysis was established, which can be observed in Figure 3.

An analysis of the main factors was performed on the model that has the best fit of the three DOE analyzed, which is modified factorial, whose results can be seen in Figure 4; which is in relation to the global desirability function, generated by the multiple optimizations of the seven mechanical properties analyzed for the composite material with natural.

## 4. Discussion

With respect to the descriptive analysis, most of the mechanical properties of the composite material have better values in the infusion molding process and the lowest percentage of variation of the data occurs when the Kevlar synthetic fiber is used, as can be seen in Table 3. The analysis data have a coefficient of variation of less than 10% in each treatment and less than 20% between treatments of each variable, in the cases that present greater variation are due to the process of generation of the hybrid material specimens especially in vacuum due to the wetting of the specimens, curing temperature; in general because the process is manual, to overcome this, advanced manufacturing processes can be used, as in the research conducted by Rajak, et al. [51]. As mentioned, very good mechanical properties of the Chambira fiber hybrid material are obtained using the infusion molding process. Thus, with this process, the highest tensile stress and elongation are generated with the kevlar reinforcement fiber whose values are in the range of 121.26 to 1311.10 MPa, and from 11.15 to 12.37% respectively; the best tensile and flexural modulus is obtained with carbon fiber reinforcement which is from 15,386.75 to 16,517.99 MPa, and from 7120.62 to 7841.21 respectively; finally with the glass reinforcement fiber a higher bending stress is obtained with a range from 71.98 to 80.26 MPa, deflection from 9.38 to 10.96 mm and with failure energy from 5.04 to 5.62 Joule, values that are higher than other researches with hybrid materials with natural fibers of similar characteristics such as flax, jute and sisal fibers in the research carried out by Benkhelladi, Laouici and Bouchoucha [62] and also in the research of Asma, Hamdi, Ali, and Youcef [63].

In the evaluation of the factorial DOE, RSM and initial Taguchi by means of ANOVA analysis, it is found that there is a difference in the means of the mechanical properties of the hybrid material in relation to the majority of the main source elements of the model as of interaction, that is, they are significant because the null statistical hypothesis (H_0_) established in Table 4 is rejected, because the *p*-values are less than 0.05 for a reliability of 95%. To improve the fit of the DOE models, modifications were made to the source elements whose means are equal with respect to the mechanical properties (H_0_ is not rejected because *p*-value is greater than 0.05), eliminating them from the analysis, thus generating more adequate DOE models in the factorial and RSM processes, on the other hand, with the Taguchi technique there is also an adequate fit when the iterations of the factors are included in the model. The above mentioned is ratified when analyzing the model parameters, it is so when comparing the standard deviation (S) values between the complete and modified models, they decrease or are maintained in each of the mechanical properties of the hybrid material analyzed; with respect to the values of the coefficient of determination R^2^ Adjusted and R^2^ of prediction, when comparing them between the initial and modified models, these improve and in their majority they are higher than 80%, similar to the research of Sinha, Bhattacharya, and Narang [37] made in composite materials with natural fiber of abaca, as well as in the research of Aly et al. [38] and from Benkhelladi, Laouici, and Bouchoucha [62] with flax, jute and sisal fibers. In addition, the variation between Adjusted and prediction R^2^ is less than 5%, which is adequate as mentioned by Mohammed, Khed, and Nuruddin [64] since it is a good indicator to obtain an appropriate model with an adequate fit. It should be noted that in the Taguchi design a prediction R^2^ value is not established, since in this technique the optimization depends on the combination of the analysis of the mean and the performance of the signal to noise ratio (S/N), for the calculation the formula of the largest category is the best was established, because it is desired to maximize the properties of the hybrid material with natural fibers of Chambira as those carried out in the research of Bezazi et al. [35] and the one by Kumar and Balachander [36] with materials composed of natural fibers.

When comparing the modified DOE models in the optimization process, which generate a better prediction R^2^ value, it is found that when analyzing individually the material properties, the RSM model obtains higher values in tensile and flexural stress and deflection; while the factorial model optimizes better the flexural modulus and failure energy; and the Taguchi model is higher in tensile modulus and elongation, as can be seen in graph 2. With respect to multiple optimization, the model that best maximizes and combines the seven mechanical properties of the composite material, is the factorial design modified by its value of the composite desirability function, since this parameter is of utmost importance to establish the quality of a prediction model as indicated by the authors Costa, Lourenco and Pereira [65], which in the present research is 0.7537 for the factorial design and is higher than the RSM of 0.7483, whose values are acceptable in relation to the research carried out by Lee, Jeong and Kim [66]. In addition, it is also found that the adjusted coefficient of determination R^2^ and prediction R^2^ of the factorial model is higher for each of the response variables in relation with the other two models analyzed.

For the three DOE analysis models, the contribution of their source elements is significant, since the ANOVA of the data generated by the composite desirability function has a *p*_value of less than 0.05. In relation to the percentage contribution of the effects globally, the most adequate model is the factorial with 99.73% which can be observed in Figure 3, that is to say that its error is 0.27% which is less than 5%, which is good in comparison with the research carried out by Baeza et al. [67] in composite materials with natural fiber in relation to the desirability function, it should be mentioned that the RSM and Taguchi models also have an error of less than 5%. With respect to the effects that have the greatest influence on the DOE models are the linear and linear iterations with two factors, as well as the quadratic effect influences the RSM.

The analysis model that presented the best results is the modified factorial, for which the surface and contour graphs shown in Figure 4 were generated, obtaining a higher desirability value with an orientation of the natural Chambira C1 fiber at 45 degrees and of the C2 fiber at 0 degrees, at a material drying temperature of 60 degrees in both cases, using the natural kevlar fiber and the Infusion forming process, taking the composite desirability value as an indicator. This can vary between 0 and 1 but not necessarily when it is closer to 1 it is established that it is good, it depends on the fulfillment of the criteria established for the optimization objective, which in the case of the present investigation is to maximize the mechanical properties of the composite material with natural fibers, as mentioned by A. Huamani et al. [68]. This confirms that the factorial model is the best and most suitable for the design of hybrid materials. By means of the combination of the indicated factors, the mechanical properties are maximized, obtaining a tensile stress in a range of (118.40; 138.10) MPa, tensile modulus of (7071.00; 8205.00) MPa, elongation of (10.77; 13.65)%, flexural stress of (69.83; 84.00) MPa, flexural modulus of (5738.00; 6459.00) MPa, deflection of (8.38; 9.60) mm and failure energy of (4.61; 5.29) Joule, with a reliability of 95%.

## 5. Conclusions

The following conclusions were drawn from the research carried out. The study reveals that the hybrid material using natural Chambira fiber together with synthetic fibers of glass, carbon and Kevlar have exceptional mechanical properties compared to other composite materials using natural fibers (such as flax, sisal and jute); the results show that tensile and flexural stress values are significantly higher. This substantial improvement in mechanical properties suggests promising potential for applications in a variety of industries, from automotive to aerospace. In addition, the analytical approach, supported by a rigorous and planned DOE design of experiments, provides a solid foundation for future interdisciplinary research in the design and development of advanced composite materials.

The research details that the factorial model stands out as the most adequate for the analysis of the mechanical properties of hybrid materials, reaching values of the adjusted coefficient of determination R^2^ and prediction R^2^ higher than 90.00%; this represents a remarkable fit in most of the mechanical properties analyzed. It is remarkable that the performance metrics of the DOE models improve by refining it, selecting only those model sources that generate difference in the means of the mechanical properties of the material. In relation to the total contribution of the model elements, the design that stands out with the best fit remains the factorial model with a value of 99.73%, which applies a linear model that incorporates interactions between the five key factors identified in this research. Alternatively, the response surface methodology, using the Box–Behnken technique continues to be effective with a contribution of 97.22%, even though it showed the lowest fit of the three models examined in relation to the R^2^ value. This approach applies a quadratic model to evaluate simple effects and linear interactions of up to two factors. The Taguchi model also presents an exceptional model contribution of 96.51% in the design of hybrid materials, employing a linear model with two-term iterations well suited to the needs of the analysis.

Combined factor optimization aimed at maximizing mechanical properties—such as tensile stress and modulus, and elongation percentage, as well as flexural stress and modulus, deflection and failure energy of the material—has demonstrated the effectiveness of the modified factorial design, as it has an overall desirability of 0.7537. With this particular model, it was found that the best combination of factors for the creation of an optimal hybrid material includes the use of natural Chambira fiber with an orientation of its plies at 45 and 0 degrees, along with synthetic Kevlar fiber, infusion processed at a drying temperature of 60 degrees Celsius. This approach not only optimizes the analysis results, but also highlights the synergy between material sciences and advanced mathematics, offering new alternatives for materials engineering research. The scientific community is invited to explore these models and contribute with innovations in this developing field, strengthening the creation of more efficient and resistant materials.

## Figures and Tables

**Figure 1 polymers-16-02051-f001:**
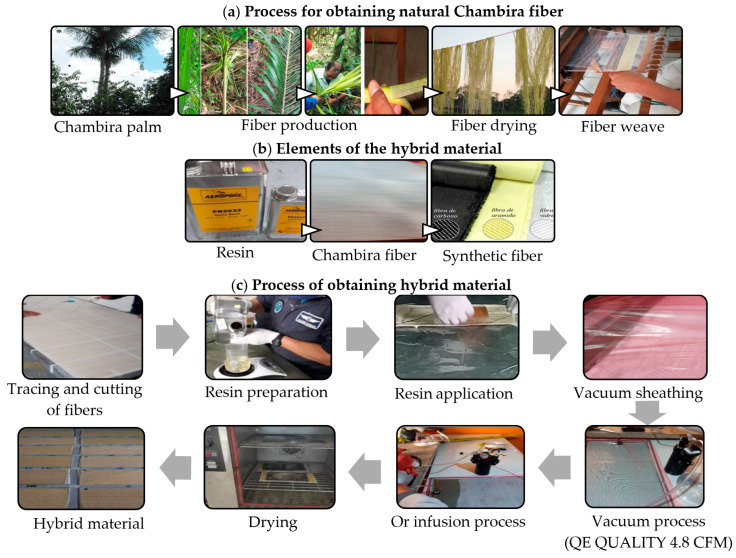
Method of manufacture of the hybrid material. In part (**a**) there is the process to obtain the natural Chambira fiber, in (**b**) the components of the hybrid material are presented and in (**c**) there is the forming process of the hybrid material of epoxy matrix reinforced with synthetic and natural fibers.

**Figure 2 polymers-16-02051-f002:**
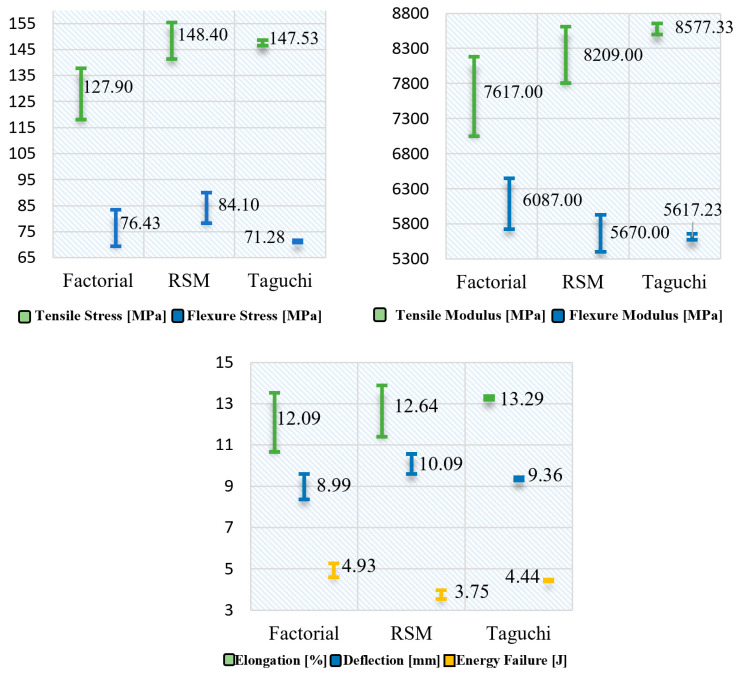
Optimization values of the mechanical properties of the composite material. The line represents confidence interval of the predicted value at 95%.

**Figure 3 polymers-16-02051-f003:**
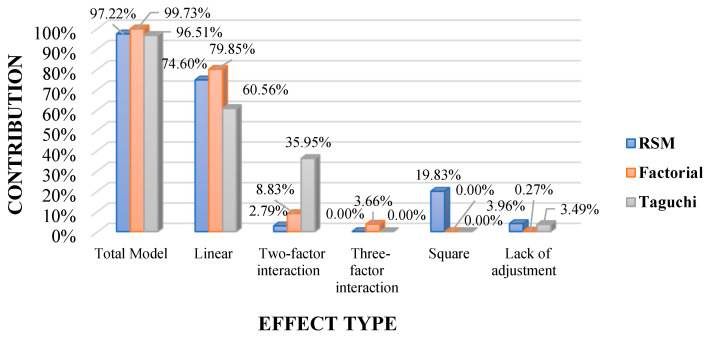
Contribution of DOE models for linear, quadratic and iteration effects. The contribution was established by the overall desirability of the modified RSM, factorial and Taguchi models.

**Figure 4 polymers-16-02051-f004:**
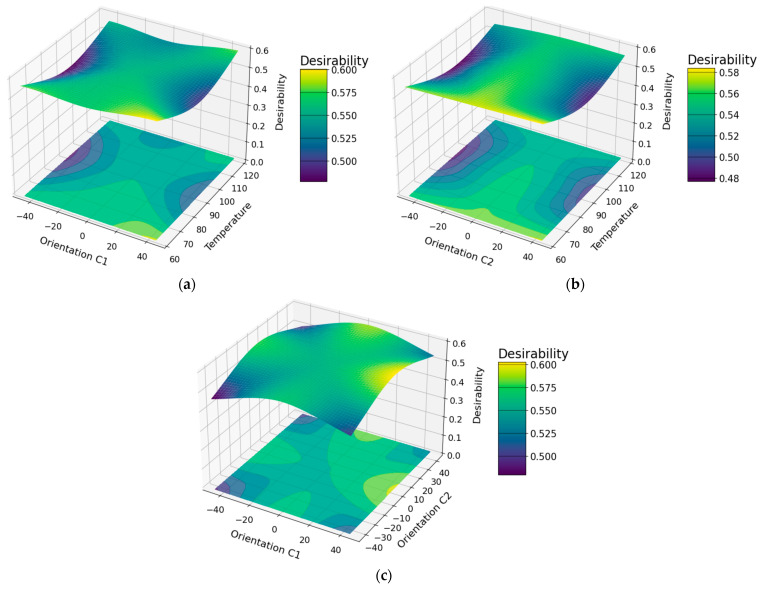
Surface and contour plots of the modified factorial model. (**a**) C1 fiber orientation, temperature and desirability, (**b**) C2 fiber orientation, temperature and desirability, and (**c**) C1, C2 fiber orientation and desirability.

**Table 1 polymers-16-02051-t001:** Applications of natural fiber hybrid.

Fiber	World Production (10^3^ ton)	Application in Building, Construction and Others
Hemp fiber	214	Construction products, textiles, cordage, geotextiles, paper and packaging, furniture, electrical, manufacture bank notes and manufacture of pipes
Flax fiber	830	Window frame, panels, decking, railing systems, fencing, tennis racket, bicycle frame, fork, seat post, snowboarding and laptop cases
Bagasse fiber	7500	Window frame, panels, decking, railing systems and fencing
Sisal fiber	375	In construction industry such as panels, doors, shutting plate and roofing sheets; also, manufacturing of paper and pulp
Kenaf fiber	970	Packing material, mobile cases, bags, insulations, clothing-grade cloth, soilless potting mixes, animal bedding, and material that absorbs oil and liquids
Coir fibers	100	Building panels, flush door shutters, roofing sheets, storage tank, packing material, helmets and postboxes, mirror casing, paper weights, projector cover, voltage stabilizer cover, a filling material for the seat upholstery, brushes and brooms, ropes and yarns for nets, bags, and mats, as well as padding for mattresses, seat cushions
Ramie fiber	100	Use in products as industrial sewing thread, packing materials, fishing nets, and filter cloths. It is also made into fabrics for household furnishings (upholstery, canvas) and clothing, paper manufacture.
Jute fiber	2300	Building panels, roofing sheets, door frames, door shutters, transport, packaging, geotextiles, and chip boards.

**Table 2 polymers-16-02051-t002:** Natural fiber properties.

Fiber	Density (g/cm^3^)	Tensile Strength (MPa)	Young’s Modulus (GPa)	Elongation at Break (%)
OPEFB	0.70–1.55	248	3.2	2.5
Linseed	1.40	88–1500	60–80	1.2–1.6
Hemp	1.48	550–900	70	1.6
Jute	1.46	400–800	10–30	1.8
Coconut fiber	1.25	220	6	15–25
Sisal	1.33	600–700	38	2–3
Abaca	1.50	980	—	—
Cotton	1.51	400	12	3–10
Kenaf (bast)	1.20	295	—	2.7–6.9
Bagasse	1.20	20–290	19.7–27.1	1.1
Henequen	1.40	430–580	—	3.0–4.7
Pineapple	1.50	170–1672	82	1–3
Banana	1.35	355	33.8	53
Cumare or Chambira	1.20	220.5	6.4	24.4

This table shows the physical and mechanical properties of natural fibers used as reinforcement for hybrid materials.

**Table 3 polymers-16-02051-t003:** Average data and variation of the mechanical properties of the hybrid material.

Property Mechanical	N. Treatment	Orientation Fiber C1 [Degrees]	Orientation Fiber C2 [Degrees]	Temp [°C]	Fiber	Process	$\mathrm{Mean}\;[\bar{x}]$	Standard Deviation [S]	Coefficient Variation [CV]
Maximum Tensile Stress [MPa]	90	−45, 0, 45	−45, 0, 45	60, 90, 120	Carbon	Empty	115.00	17.11	14.88%
						Infusion	103.66	16.52	15.94%
					Glass	Empty	81.37	13.32	16.37%
						Infusion	79.52	11.64	14.64%
					Kevlar	Empty	129.33	17.73	13.71%
						Infusion	126.18	16.85	13.35%
Modulus Elasticity Tensile [MPa]	90	−45, 0, 45	−45, 0, 45	60, 90, 120	Carbon	Empty	9999.89	1580.55	15.81%
						Infusion	15,952.37	1935.92	12.14%
					Glass	Empty	5632.29	1025.51	18.21%
						Infusion	5964.37	859.13	14.40%
					Kevlar	Empty	13,441.81	2124.36	15.80%
						Infusion	7044.92	1191.75	16.92%
Elongation [%]	90	−45, 0, 45	−45, 0, 45	60, 90, 120	Carbon	Empty	6.02	1.03	17.19%
						Infusion	3.14	0.57	18.20%
					Glass	Empty	7.20	1.17	16.18%
						Infusion	8.41	1.59	18.90%
					Kevlar	Empty	6.77	1.13	16.70%
						Infusion	11.76	2.09	17.80%
Maximum Flexure Stress [MPa]	90	−45, 0, 45	−45, 0, 45	60, 90, 120	Carbon	Empty	49.48	8.51	17.19%
						Infusion	67.90	13.51	19.90%
					Glass	Empty	69.80	13.55	19.42%
						Infusion	76.12	14.16	18.61%
					Kevlar	Empty	65.06	11.37	17.48%
						Infusion	68.67	12.22	17.80%
Modulus Elasticity Flexure [MPa]	90	−45, 0, 45	−45, 0, 45	60, 90, 120	Carbon	Empty	3238.33	631.98	19.52%
						Infusion	7480.91	1233.15	16.48%
					Glass	Empty	2920.62	578.90	19.82%
						Infusion	5170.04	747.67	14.46%
					Kevlar	Empty	2683.98	428.80	15.98%
						Infusion	5018.09	847.50	16.89%
Deflection [mm]	90	−45, 0, 45	−45, 0, 45	60, 90, 120	Carbon	Empty	7.37	1.43	19.44%
						Infusion	9.30	1.71	18.42%
					Glass	Empty	7.73	1.40	18.17%
						Infusion	10.39	1.94	18.67%
					Kevlar	Empty	7.98	1.32	16.53%
						Infusion	10.09	1.93	19.11%
Energy Failure [J]	90	−45, 0, 45	−45, 0, 45	60, 90, 120	Carbon	Empty	2.79	0.55	19.62%
						Infusion	4.26	0.81	18.91%
					Glass	Empty	5.18	0.98	18.91%
						Infusion	5.33	0.99	18.61%
					Kevlar	Empty	4.38	0.81	18.46%
						Infusion	4.38	0.81	18.46%

**Table 4 polymers-16-02051-t004:** Hypotheses for the evaluation of design of experiments models.

DOE Model Source	Hypothesis	Description
Molding process	H_0_	The average mechanical properties with the vacuum casting and infusion process are similar.
	H_1_	The average mechanical properties with the vacuum casting and infusion process are different.
Synthetic fiber type	H_0_	All averages of mechanical properties with the synthetic fiber types glass, carbon and Kevlar are similar.
	H_1_	At least one of the pairs of average mechanical properties with the synthetic fiber types glass, carbon and Kevlar is different.
Orientation FCh. C1 *	H_0_	All mechanical property averages with the orientations of the first layer of Chambira natural fiber at −45°, 0° and 45° are similar.
	H_1_	At least one of the pairs of average mechanical properties with the orientations of the first layer of Chambira natural fiber at −45°, 0° and 45° is different.
Orientation FCh. C2 *	H_0_	All mechanical property averages with the second layer orientations of Chambira natural fiber at −45°, 0° and 45° are similar.
	H_1_	At least one of the pairs of average mechanical properties with the orientations of the second layer of Chambira natural fiber at −45°, 0° and 45° is different.
Temperature	H_0_	All mechanical property averages at the drying temperatures of 60 °C, 90 °C and 120 °C are similar.
	H_1_	At least one of the pairs of average mechanical properties at drying temperatures of 60 °C, 90 °C and 120 °C is different.
Linear and quadratic iterations	H_0_	All mechanical property means with linear and quadratic iterations of the model sources are similar.
	H_1_	At least one of the pairs in averages of the mechanical properties with linear and quadratic iterations of the model sources is different.

* FCh: Chambira Fiber, C1: Layer 1, C2: Layer 2.

**Table 5 polymers-16-02051-t005:** ANOVA of the full factorial model for the mechanical properties of the hybrid material.

ANOVA	*p*-Value						
Full Factor Model Source	Tensile Stress	Modulus Elasticity Tensile	Elongation	Flexure Stress	Modulus Elasticity Flexure	Deflection	Energy Failure
Model	0.000	0.000	0.000	0.000	0.000	0.000	0.000
Linear	0.000	0.000	0.000	0.000	0.000	0.000	0.000
Orientation FCh * C1	0.000	0.000	0.000	0.000	0.003	0.015	0.000
Orientation FCh * C2	0.000	0.000	0.000	0.000	0.000	0.000	0.000
Temperature	0.000	**0.093**	0.000	0.000	0.002	0.000	0.000
Moldeo	0.000	0.003	0.000	0.000	0.000	0.000	0.000
Synthetic Fiber	0.000	0.000	0.000	0.000	0.000	0.000	0.000
Two-term interactions	0.000	0.000	0.000	0.000	0.000	0.000	0.000
Orient. FCh * C1_Moldeo	**0.411**	**0.745**	0.032	0.000	**0.752**	0.000	**0.579**
Orient. FCh * C1_Synthetic Fiber	0.000	0.025	0.000	0.002	0.012	0.000	0.000
Orient. FCh * C2_Moldeo	**0.479**	0.014	**0.105**	0.000	0.040	0.018	**0.606**
Orient. FCh * C2_ Synthetic Fiber	0.014	**0.071**	0.000	0.000	0.009	0.000	0.000
Temperature_Moldeo	**0.414**	0.007	**0.086**	0.000	0.000	0.000	**0.637**
Temperature_Synthetic Fiber	0.000	0.000	0.000	0.000	0.000	0.000	0.000
Moldeo_Synthetic Fiber	0.044	0.000	0.000	0.000	0.000	0.001	0.000
Three-term interactions	0.002	0.007	0.000	0.000	0.000	0.000	**0.966**
Orient. C1_Moldeo_Synthetic Fiber	**0.557**	**0.719**	**0.063**	0.006	**0.590**	0.000	**0.793**
Orient. C2_Moldeo_Synthetic Fiber	0.050	0.006	**0.163**	**0.068**	0.000	**0.286**	**0.732**
Temp_Moldeo_Synthetic Fiber	0.001	0.023	0.000	0.000	0.000	0.000	**0.808**
Normality (Kolmogorov-Smirnov)	**0.060**	**0.150**	**0.150**	**0.074**	**0.128**	**0.080**	0.005
Full Factor Model Parameters							
S (Standard Deviation)	0.10	0.10	0.02	0.01	0.08	0.77	0.02
R^2^ (%)	85.26	95.49	94.75	81.18	96.81	87.47	91.52
R^2^ Adjusted (%)	82.60	94.68	93.81	77.79	96.24	85.22	89.99
R^2^ Prediction (%)	79.16	93.76	92.53	73.33	95.56	82.43	88.13

* FCh: Chambira fiber, bold values indicate *p* > 0.05.

**Table 6 polymers-16-02051-t006:** ANOVA of the modified factorial model for the mechanical properties of the hybrid material.

ANOVA	*p*-Value						
Modified Factor Model Source	Tensile Stress	Modulus Elasticity Tensile	Elongation	Flexure Stress	Modulus Elasticity Flexure	Deflection	Energy Failure
Model	0.000	0.000	0.000	0.000	0.000	0.000	0.000
Linear	0.000	0.000	0.000	0.000	0.000	0.000	0.000
Orientation FCh * C1	0.000	0.000	0.000	0.000	0.003	0.016	0.000
Orientation FCh * C2	0.000	0.000	0.000	0.000	0.000	0.000	0.000
Temperature	0.000	**0.091**	0.000	0.000	0.002	0.000	0.000
Moldeo	0.000	0.002	0.000	0.000	0.000	0.000	0.000
Synthetic Fiber	0.000	0.000	0.000	0.000	0.000	0.000	0.000
Two-term interactions	0.000	0.000	0.000	0.000	0.000	0.000	0.000
Orient. FCh * C1_Moldeo	**0.418**	**0.743**	0.035	0.000	**0.751**	0.000	**0.570**
Orient. FCh * C1_Synthetic Fiber	0.000	0.024	0.000	0.003	0.011	0.000	0.000
Orient. FCh * C2_Moldeo	**0.485**	0.014	0.112	0.000	0.039	0.018	**0.596**
Orient. FCh * C2_ Synthetic Fiber	0.015	**0.069**	0.000	0.000	0.009	0.000	0.000
Temperature_Moldeo	**0.420**	0.006	0.092	0.000	0.000	0.000	**0.628**
Temperature_Synthetic Fiber	0.000	0.000	0.000	0.000	0.000	0.000	0.000
Moldeo_Synthetic Fiber	**0.103**	0.000	0.000	0.000	0.000	0.001	0.000
Three-term interactions	0.001	0.001	0.000	0.000	0.000	0.000	-
Orient. C1_Moldeo_Synthetic Fiber	-	0.007	-	0.005	-	0.000	-
Orient. C2_Moldeo_Synthetic Fiber	-	-	-	-	0.000	-	-
Temp_Moldeo_Synthetic Fiber	0.001	0.021	0.000	0.000	0.000	0.000	-
Normality (Kolmogorov-Smirnov)	**0.078**	**0.150**	**0.071**	**0.074**	**0.128**	**0.150**	0.05
Modified Factor Model Parameters							
S (Standard Deviation)	0.11	0.10	0.02	0.01	0.08	0.77	0.02
R^2^ (%)	84.47	95.45	94.41	80.45	96.77	87.20	91.34
R^2^ Adjusted (%)	82.29	94.73	93.62	77.33	96.26	85.15	90.30
R^2^ Prediction (%)	79.59	93.90	92.58	73.43	95.67	82.66	89.18

* FCh: Chambira fiber, bold values indicate *p* > 0.05.

**Table 7 polymers-16-02051-t007:** ANOVA of the Taguchi model for the mechanical properties of the hybrid material.

ANOVA	*p*-Value						
Taguchi Model Source without Iterations	Tensile Stress	Modulus Elasticity Tensile	Elongation	Flexure Stress	Modulus Elasticity Flexure	Deflection	Energy Failure
Orientation FCh * C1	0.001	**0.267**	**0.748**	**0.330**	**0.789**	**0.548**	**0.058**
Orientation FCh * C2	0.000	**0.132**	**0.601**	0.000	0.000	0.000	0.227
Temperature	**0.084**	**0.878**	**0.731**	0.032	**0.947**	0.011	0.007
Moldeo	0.031	**0.916**	0.032	0.000	0.000	0.000	0.005
Synthetic Fiber	0.000	0.000	0.000	0.000	0.000	0.039	0.000
Normality (Kolmogorov-Smirnov)	**0.150**	**0.057**	**0.150**	**0.144**	**0.150**	**0.150**	**0.099**
Taguchi Model parameters without iterations							
S (Standard Deviation)	11.76	2860.81	2.18	10.41	738.02	1.10	0.84
R^2^ (%)	78.54	76.10	75.49	73.18	85.40	74.53	73.27
R^2^ Adjusted (%)	75.69	70.29	71.68	71.98	83.47	71.16	71.09

* FCh: Chambira fiber, bold values indicate *p* > 0.05.

**Table 8 polymers-16-02051-t008:** ANOVA of the modified Taguchi model for the mechanical properties of the hybrid material.

ANOVA	*p*-Value						
Taguchi Model Source with Iterations	Tensile Stress	Modulus Elasticity Tensile	Elongation	Flexure Stress	Modulus Elasticity Flexure	Deflection	Energy Failure
Orientation FCh * C1	0.002	0.000	**0.386**	**0.305**	**0.602**	**0.457**	0.001
Orientation FCh * C2	0.000	0.000	**0.192**	0.000	0.000	0.000	0.019
Temperature	**0.096**	**0.268**	**0.357**	0.027	**0.889**	0.004	0.000
Moldeo	0.035	**0.678**	0.000	0.000	0.000	0.000	0.000
Synthetic Fiber	0.000	0.000	0.000	0.000	0.000	0.024	0.000
Orient. FCh * C1_Moldeo	**0.916**	**0.901**	**0.639**	**0.147**	**0.819**	**0.338**	**0.954**
Orient. FCh * C1_Synthetic Fiber	**0.470**	**0.824**	**0.188**	**0.567**	**0.845**	**0.281**	0.000
Orient. FCh * C2_Moldeo	**0.807**	**0.176**	**0.844**	**0.200**	**0.158**	**0.550**	**0.922**
Orient. FCh * C2_ Synthetic Fiber	**0.870**	**0.707**	**0.203**	**0.550**	**0.315**	**0.530**	0.001
Temperature_Moldeo	**0.638**	0.030	**0.350**	0.034	0.006	0.000	**0.950**
Temperature_Synthetic Fiber	**0.193**	0.005	0.021	**0.563**	**0.058**	**0.130**	0.000
Moldeo_Synthetic Fiber	**0.411**	0.000	0.000	**0.095**	0.000	**0.311**	0.000
Normality (Kolmogorov-Smirnov)	**0.150**	**0.150**	**0.061**	**0.150**	**0.150**	**0.150**	0.050
Taguchi Model Parameters with Iterations							
S (Standard Deviation)	11.99	889.30	1.19	10.01	503.37	0.959	0.50
R^2^ (%)	84.24	97.01	89.10	82.42	95.20	86.32	84.34
R^2^ Adjusted (%)	77.71	95.20	85.52	80.95	92.31	78.06	81.30

* FCh: Chambira fiber, bold values indicate *p* > 0.05.

**Table 9 polymers-16-02051-t009:** ANOVA of the initial RSM model for the mechanical properties of the hybrid material.

ANOVA	*p*-Value						
Initial RSM Model Source	Tensile Stress	Modulus Elasticity Tensile	Elongation	Flexure Stress	Modulus Elasticity Flexure	Deflection	Energy Failure
Model	0.000	0.000	0.000	0.000	0.000	0.000	0.000
Linear	0.000	0.000	0.000	0.000	0.000	0.000	0.000
Orientation FCh * C1	**0.486**	**0.446**	**0.455**	**0.202**	0.031	**0.081**	0.018
Orientation FCh *C2	**0.350**	**0.151**	**0.606**	**0.448**	**0.477**	0.000	**0.640**
Temperature	0.001	**0.114**	0.000	0.000	**0.067**	0.000	0.014
Moldeo	0.001	0.005	0.001	0.000	0.000	0.000	0.000
Synthetic Fiber	0.000	0.000	0.000	0.000	0.000	0.000	0.000
Square	0.000	0.000	0.000	0.000	0.000	0.000	0.000
Orient. FCh * C1_Orient.FCh * C1	0.000	0.000	0.002	0.001	**0.106**	**0.199**	0.000
Orient. FCh * C2_Orient. FCh * C2	0.000	0.000	0.000	0.000	0.000	0.000	0.000
Temperature _Temperature	0.078	**0.220**	**0.915**	**0.441**	0.027	0.000	0.000
Two-term interactions	**0.155**	0.000	0.000	0.000	0.000	0.000	0.000
Orient. FCh * C1_Orient. FCh * C2	**0.535**	0.002	**0.675**	0.000	**0.408**	**0.177**	0.043
Orient. FCh * C1_Temperature	**0.645**	**0.930**	**0.057**	**0.573**	**0.566**	**0.856**	0.014
Orient. FCh * C1_Moldeo	**0.310**	**0.995**	**0.052**	**0.095**	**0.808**	**0.790**	**0.976**
Orient. FCh * C1_ Synthetic Fiber	**0.801**	**0.563**	**0.667**	**0.136**	**0.471**	**0.320**	0.003
Orient. FCh * C2_Temperature	**0.637**	**0.119**	0.001	**0.177**	**0.461**	**0.584**	**0.157**
Orient. FCh * C2_Moldeo	**0.725**	**0.062**	**0.250**	0.036	**0.930**	**0.385**	**0.745**
Orient. FCh * C2_Synthetic Fiber	**0.127**	**0.482**	**0.771**	0.003	**0.543**	**0.379**	0.000
Temperature_Moldeo	**0.255**	**0.056**	**0.117**	0.000	0.000	0.000	**0.951**
Temperature_Synthetic Fiber	**0.116**	0.029	0.000	0.044	0.000	0.000	0.000
Moldeo_Synthetic Fiber	0.027	0.000	0.000	0.000	0.000	0.049	0.000
Normality (Kolmogorov-Smirnov)	**0.056**	**0.150**	**0.150**	**0.150**	**0.069**	**0.053**	0.005
Full RSM Model Parameters							
S (Standard Deviation)	0.12	0.11	0.03	0.01	0.99	1.03	0.03
R^2^ (%)	85.35	94.19	89.70	83.22	94.53	75.64	82.85
R^2^ Adjusted (%)	81.33	93.65	88.73	80.22	94.02	73.36	81.24
R^2^ Prediction (%)	78.64	93.06	87.52	76.00	93.39	70.88	79.87

* FCh: Chambira fiber, bold values indicate *p* > 0.05.

**Table 10 polymers-16-02051-t010:** ANOVA of the modified RSM Model for the mechanical properties of the hybrid material.

ANOVA	*p*-Value						
Modified RSM Model Source	Tensile Stress	Modulus Elasticity Tensile	Elongation	Flexure Stress	Modulus Elasticity Flexure	Deflection	Energy Failure
Model	0.000	0.000	0.000	0.000	0.000	0.000	0.000
Linear	0.000	0.000	0.000	0.000	0.000	0.000	0.000
Orientation FCh * C1	**0.488**	**0.449**	**0.457**	**0.205**	0.030	-	0.017
Orientation FCh * C2	**0.353**	**0.154**	**0.608**	**0.451**	**0.475**	0.000	**0.638**
Temperature	0.001	**0.117**	0.000	0.000	**0.065**	0.000	0.013
Moldeo	0.001	0.005	0.001	0.000	0.000	0.000	0.000
Synthetic Fiber	0.000	0.000	0.000	0.000	0.000	0.000	0.000
Square	0.000	0.000	0.000	0.000	0.000	0.000	0.000
Orient. FCh * C1_Orient. FCh * C1	0.000	0.000	0.002	0.001	-	-	0.000
Orient. FCh * C2_Orient. FCh * C2	0.000	0.000	0.000	0.000	0.000	0.000	0.000
Temperature _Temperature	-	-	-	-	0.018	0.000	0.000
Two-term interactions	0.028	0.000	0.000	0.000	0.000	0.000	0.000
Orient. FCh * C1_Orient. FCh * C2	-	0.002	0.001	0.000	-	-	0.042
Orient. FCh * C1_Temperature	-	-	-	-	-	-	0.014
Orient. FCh * C1_ Synthetic Fiber	-	-	-	-	-	-	0.003
Orient. FCh * C2_Temperature	-	-	-	-	-	-	0.000
Orient. FCh * C2_Moldeo	-	-	-	0.037	-	-	-
Orient. FCh * C2_Synthetic Fiber	-	-	-	0.004	-	-	-
Temperature_Moldeo	-	-	-	0.000	0.000	0.000	-
Temperature_Synthetic Fiber	-	0.031	0.000	0.046	0.000	0.000	0.000
Moldeo_Synthetic Fiber	0.028	0.000	0.000	0.000	0.000	0.049	0.000
Normality (Kolmogorov-Smirnov)	**0.052**	**0.150**	**0.150**	**0.150**	**0.051**	**0.150**	0.005
Modified MSR Model Parameters							
S (Standard Deviation)	0.12	0.11	0.03	0.01	0,10	1.04	0.03
R^2^ (%)	83.00	93.87	89.16	81.90	94.38	74.45	82.70
R^2^ Adjusted (%)	81.11	93.56	88.61	79.74	94.09	73.26	81.38
R^2^ Prediction (%)	79.96	93.20	87.86	77.17	93.77	71.81	80.23

* FCh: Chambira fiber, bold values indicate *p* > 0.05.

## Data Availability

The data presented in this study are available on request from the corresponding author due to privacy and ethical reasons.

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
