# Peer review of "Validation of DOE Factorial/Taguchi/Surface Response Models of Mechanical Properties of Synthetic and Natural Fiber Reinforced Epoxy Matrix Hybrid Material"

_polymers, 2024, doi:10.3390/polym16142051_

Round 1
Reviewer 1 Report (Previous Reviewer 3)
Comments and Suggestions for Authors
My main concerns were addressed. The paper can be accepted as is.
Author Response
For review article
|
Response to Reviewer 1 Comments |
|
1. Summary |
|
|
|
Sincere thanks for your time in reviewing this manuscript. In the forwarded files you will find my detailed responses, along with highlighted revisions, corrections and changes made. |
||
|
|
|
|
|
2. Questions for General Evaluation |
Reviewer’s Evaluation |
Response and Revisions |
|
Is the work a significant contribution to the field? |
|
The corresponding changes were made. |
|
Is the work well organized and comprehensively described? |
|
|
|
Is the work scientifically sound and not misleading? |
|
|
|
Are there appropriate and adequate references to related and previous work? |
|
|
|
Is the English used correct and readable? |
|
|
|
3. Point-by-point response to Comments and Suggestions for Authors |
||
Comments 1: My main concerns are addressed. The document can be accepted as is.
Response 1: Thank you for reviewing and accepting the document, but because it is marked in the evaluation review that it should be improved and there is no comment that mentions how this improvement should be made. It should be mentioned that all the changes previously requested were made, as well as the new minor observations made by reviewer 2 and the academic reviewer, which are the improvement in the size of figure 1, change in the numbering of subtitle 3.2. It should also be noted that the Minitab program version 21 was used for the analysis of the results, Microsoft Excel was used for figures 2 and 3, and Python software version 3.11 was used for figure 4.
- Additional clarifications
The changes are shown in red in the article document.

Reviewer 2 Report (Previous Reviewer 1)
Comments and Suggestions for Authors
Although great changes have been made, several issues should still be addressed.
1. Figure 1 should be carefully edited. The present version is too big.
2. The subtitle of "3.1. Optimization of the mechanical properties of hybrid material " should be corrected.
3. Figure 2 was made using what software?
Comments on the Quality of English LanguageProper modification should be performed.
Author Response
For review article
|
Response to Reviewer 2 Comments |
|||
|
1. Summary |
|
|
|
|
Sincere thanks for your time in reviewing this manuscript. In the forwarded files you will find my detailed responses, along with highlighted revisions, corrections and changes made. |
|||
|
2. Questions for General Evaluation |
Reviewer’s Evaluation |
Response and Revisions |
|
|
Is the work a significant contribution to the field? |
|
The corresponding changes detailed in the answers to each item were made. |
|
|
Is the work well organized and comprehensively described? |
|
|
|
|
Is the work scientifically sound and not misleading? |
|
|
|
|
Are there appropriate and adequate references to related and previous work? |
|
|
|
|
Is the English used correct and readable? |
|
|
|
|
3. Point-by-point response to Comments and Suggestions for Authors |
|||
Comments 1: Figure 1 should be edited carefully. The current version is too large.
Response 1: Thank you for pointing this out. We have revised and improved Figure 1 with a more appropriate size, which you have on page 4, paragraph 7 and line 132.
Comments 2: The subtitle of "3.1. Optimization of the mechanical properties of the hybrid material" should be corrected.
Response 2: The reviewer's suggestion is accepted, we have revised and corrected the subtitle, changing the numbering from 3.1 to 3.2 because it is the correct one, the change is on page 11, paragraph 21 and line 295.
Comments 3: ¿Using which software was figure 2 made?
Response 3: Figure 2 was made with the Microsoft Excel program with a bar chart, using the averages of the predicted values for the optimization of the mechanical properties and their 95% confidence intervals. It should be noted that for a better observation of the confidence interval, the bars are excluded. For the analysis of the results, the Minitab version 21 program was used, for figures 2 and 3, as mentioned, the Microsoft Excel program was used and for figure 4, the Python version 3.11 software was used.
- Additional clarifications
The changes are shown in red in the article document.

This manuscript is a resubmission of an earlier submission. The following is a list of the peer review reports and author responses from that submission.
Round 1
Reviewer 1 Report
Comments and Suggestions for Authors
In this work, Toapanta et al reported the validation of DOE factorial/taguchi/surface response models of mechanical properties of synthetic and natural fiber rein-forced epoxy based hybrid material. There are several points that should be addressed before the article may be considered for publication.
1. The format of the manuscript should be further corrected. For example, “methodology [41], [42]On the influence of the orientation ---”
2. The references should be cited properly. For example, “which are modified by chemical treatment [29][30], [31], [32], [33]” may be exhibited as “which are modified by chemical treatment [29-33]”.
3. The color of the inset words in Fig.1 should be changed. In the present version, it is hard to recognize them.
4. It is not good putting references after the Figure or table captions. For example, the references in “Table 2. Natural fiber properties [3], [28]” should be removed.
5. Capital letter should be properly used. For example, “such as Glass Fiber”.
Reviewer 2 Report
Comments and Suggestions for Authors
This work examines the use of three experimental design models—Factorial, Taguchi, and Response Surface Methodology —to evaluate and optimise natural fibre-reinforced composite materials. The modified Factorial Design maximises the combination of mechanical properties within the composite material. It offers an practical approach for predicting the mechanical properties of natural fibre-reinforced composite materials. This manuscript can be accepted when the following issues have been addressed:
1. In the introduction, the author should emphasize the unique aspects and innovations of this research in comparison to other studies.
2. The experimental details about the fabrication of the composite materials need to be clearer. The images in Figure 1 are confusing for readers.
3. The author should consistently use either periods or commas when presenting numerical values.
Comments on the Quality of English LanguageMinor editing of English language required
Reviewer 3 Report
Comments and Suggestions for Authors
Abstract:
1. Clarify model validation steps; details are unclear.
2. 15 cases are too few for reliable conclusions.
3. R² > 70% not consistently significant; re-evaluate criteria.
4. Explain "greater than 87%" contribution; lacks context.
5. 0.4442 and 0.4387 indicate weak model performance; reconsider methodology.
6. Quantify "better prediction" with modified Factorial model precisely.
Section 1:
1. 50+ references in the introduction; keep only to key studies.
2. Table 1 lacks specific quantitative performance metrics.
3. Clarify chemical treatment methods and impacts.
4. "Several advantages" vague; specify mechanical benefits.
5. Provide detailed steps for Factorial, Taguchi, RSM.
6. Clearly state research hypothesis and objectives.
7. Consider citing https://doi.org/10.25518/esaform21.4743
Section 2:
1. Quasi-experimental design lacks direct variable manipulation; consider true experimental.
2. Non-probabilistic sampling limits generalizability; justify choice quantitatively.
3. Only 15 cases analyzed; increase sample size for robustness.
4. Fiber orientation levels (-45, 0, 45 degrees) need justification.
5. Synthetic fiber types lack detailed selection criteria; clarify rationale.
6. Explain calculation and selection method for 270 specimens.
Section 3:
1. Coefficient of Variation > 30% unreliable; investigate sources.
2. Significant p-value variability across models; clarify inconsistencies.
3. 15 cases per combination inadequate for robust conclusions.
4. R² ranges from 44.96% to 92.69%; address model accuracy.
5. Provide λ values for each Box-Cox transformation.
Section 4:
1. Inconsistent tensile stress range (80.58-128.52 MPa); clarify variation sources.
2. Taguchi model not significant for stress/modulus; review methodology.
3. High standard deviations, e.g., 20.25; investigate variation sources.